# A New 3-D Open-Framework Zinc Borovanadate with Catalytic Potentials in α-Phenethyl Alcohol Oxidation

**DOI:** 10.3390/molecules24030531

**Published:** 2019-02-01

**Authors:** Xinxin Liu, Biao Guo, Xuejiao Sun, Le Zhang, Hongming Yuan

**Affiliations:** 1State Key Laboratory of Inorganic Synthesis and Preparative Chemistry, College of Chemistry, Jilin University, Changchun 130012, China; liuxinxin1114@163.com (X.L.); sunxj17@jlu.edu.cn (X.S.); zhangle16@jlu.edu.cn (L.Z.); 2Institute of Catalysis for Energy and Environment, College of Chemistry and Chemical Engineering, Shenyang Normal University, Shenyang 110034, China; biaoguo14@126.com

**Keywords:** borovanadate, open framework, three-dimensional, catalytic activities

## Abstract

A novel 3-D open-framework zinc borovanadate [Zn_6_(en)_3_][(V^IV^O)_6_(V^V^O)_6_O_6_(B_18_O_36_(OH)_6_)·(H_2_O)]_2_·14H_2_O (**1**, en = ethylenediamine) was hydrothermally obtained and structurally characterized. The framework was built from [V_12_B_18_O_54_(OH)_6_(H_2_O)]^10−^ polyanion clusters bridged by Zn(en) complex fragments. The compound not only possessed a three-dimensional open-framework structure with unique plane-shaped channels, but also exhibited excellent catalytic activities for the oxidation of α-phenethyl alcohol.

## 1. Introduction

Research on inorganic framework solids possessing open architectures continues to attract great interest due to their many potential applications [1,2,3,4,5,6]. Inorganic open-framework solids with large channels or cavities have been developed for adsorption and catalysis studies [7,8]. It is becoming equally important to search for new framework solids with attractive structures and promising properties [9,10,11]. To date, almost half of the elements on the periodic table can be used to construct open-framework materials. The scope of such frameworks is no longer confined to traditional silicates [12], phosphates [13], etc., but also includes emerging borates and their derivatives, such as borogermanates [14], borophosphates [15], and borovanadates [16]. 

In recent years, borovanadates have attracted extensive attention in solid-state material chemistry owing to their fascinating architectures and wide applications in adsorption [7], catalysis [17], and magnetism [18]. A vast range of applications has led to great efforts toward rationally designing and synthesizing such compounds. Until now, various BVO clusters have been observed, as exemplified by V_6_B_20_ in [(VO)_6_{B_10_O_16_(OH)_6_}_2_]^3−^ [19], V_6_B_22_ in [V_6_B_22_O_44_(OH)_10_]^8−^ [20], V_10_B_28_ in [Mn_4_(C_2_O_4_)(V_10_B_28_O_74_H_8_)]^10−^ [21], V_12_B_16_ in [(VO)_12_O_4_{B_8_O_17_(OH)_4_}_2_]^8−^ [22,23], V_12_B_17_ in [(VO)_12_B_17_O_38_(OH)_8_}^9−^ [24], V_12_B_18_ in [(VO)_12_O_6_{B_3_O_6_(OH)}_6_]^10−^ [24], and V_12_B_32_ in [(VO)_12_{B_16_O_32_(OH)_4_}_2_]^12−^ [23,25]. As is well known, borovanadate anions are a class of discrete transition metal oxide clusters with high negative charges and oxygen-rich surfaces. These anions are one of the most ideal multi-dentate nucleophilic candidates for the construction of extended open frameworks [26]. Therefore, an approach that is widely used for designing such open frameworks is to link borovanadate polyanion clusters with different types of linkers of transition metal (TM) ions [21] or TM complex moieties [27]. For example, in 2012, Zhou et al. addressed a Na borovanadate with a 1-D framework [28]. In 2015, Han et al. reported a 2-D Ni borovanadate [Ni(en)_2_]_6_[(VO)_12_O_6_B_18_O_3_(OH)_3_]·5H_2_O by using ethylenediamine (en) as a template [29]. Although many extended borovanadates have now been reported [22,30,31], relatively little progress has been made in the preparation of three-dimensional open-framework borovanadates, which are known thus far only for SUT-6 [17], SUT-7-Zn [7], {[Cu(dien)(H_2_O)]_3_V_12_B_18_O_54_(OH)_6_(H_2_O)}·4H_3_O·5.5H_2_O [32], etc., (as listed in Appendix A), likely due to the difficulty of controlling the terminal or bridging oxygen atoms of polyanions that bond bridging transition metal complex moieties. Thus, the preparation of 3-D open-framework polyoxoborovanadates based on the linkages of borovanadate polyanions and TM/TM complex bridges still remains challenging.

Herein, we hydrothermally synthesized a new 3-D open-framework borovanadate (**1**), which was constructed by the linkages of Zn(en) complex fragments and [V_12_B_18_] clusters. This process resulted in a rare example of extended borovanadate with unique plane-shaped channels. This compound not only enriches the family of open-framework borovanadates, but also displays high catalytic activities for the oxidation of α-phenethyl alcohol.

## 2. Results and Discussion

### 2.1. Synthetic Aspects and Characterization

The preparation of compound **1** was conducted at 180 °C for seven days using a hydrothermal method. When the temperature declined or reaction time was reduced, crystals could not be obtained. The number of amines used in the synthesis had a significant effect on the dimensionality of the resulting vanadoborate structures. Compound **1** was only obtained when the amount of ethanediamine and NH_4_VO_3_ was in a molar ratio of 10:1. In our reaction conditions, when the amount of ethanediamine was between 0.5 and 5 mmol, only amorphous products were obtained. When the amount of ethanediamine varied from 6 to 9 mmol, a 0-D cage structure [Zn(en)_2_]_6_[(VO)_12_O_6_B_18_O_39_(OH)_3_]·13H_2_O [33] was formed, together with black impurities. When the amount of ethanediamine was increased to 10 mmol, pure compound **1** could be synthesized. When the amount of ethanediamine was further increased to 11 mmol, a mixture of compound **1** with poor crystallinity and brown block impurities was obtained. Otherwise, in order to further investigate the influence of the distinct metal ions on the structural construction of 3-D products, Mn(CH_3_COO)_2_·4H_2_O, Cu(NO_3_)_2_·3H_2_O, and Ni(NO_3_)_2_·6H_2_O were used instead of Zn(NO_3_)_2_·6H_2_O. However, isotypic compounds could not be obtained. This result demonstrates that Zn^2+^ ions played a pivotal structure-directing role in the formation of the open framework.

As shown in Figure 1, the experimental powder X-ray diffraction pattern is consistent with the simulated one originating from single-crystal structural data, indicating the phase purity of the as-synthesized product. The result of the thermogravimetric analyses is shown in Appendix A. The thermogravimetric (TG) analysis curve shows a weight loss of 5.60% between room temperature and 140 °C, which is the calculated value for 14 water molecules. The successive weight loss of 10.84% from 140 to 420 °C can be assigned to the release of three ethylenediamine (en) ligands, two central water molecules, and twelve -OH groups formed in the {(VO)_6_[B_10_O_16_(OH)_6_]_2_}^3−^ polyanion (calc. 9.60%). Subsequently, gradual weight loss above 420 °C may be attributed to oxygen release by the decomposition of V_2_O_5_.

In the IR spectrum (Appendix A), the absorption bands from 3400 to 3000 cm^−1^ are due to O–H and N–H stretching vibration. The bands at about 1621 and 1517 cm^−1^ are assigned to O–H and N–H bending vibration. The band at 1385 cm^−1^ is ascribed to the B–O asymmetric stretching of [BO_3_] units, whereas that of the [BO_4_] units appears at 1054 cm^−1^. The bands at 926–942 cm^−1^ are assigned to the terminal V–O asymmetrical stretching. A series of bands in the 662–791 cm^−1^ region can be attributed to symmetrical and asymmetrical V–O–V stretching [34,35]. The V–O–B asymmetric stretching is located at 521 cm^−1^ [36]. Appendix A presents the UV-Vis-NIR spectrum of compound **1**. The absorption band centered at 550 nm presumably arises from an intervalence charge transfer transition (IVCT) between the V^IV^/V^V^ centers of the mixed valence vanadoborate unit. The absorptions in the high-energy UV region at 224 and 350 nm can be assigned to the O→V and O→B charge transfers of the [V_12_B_18_O_54_(OH)_6_(H_2_O)]^10−^ cluster [26,37].

### 2.2. Structure Description

The single-crystal X-ray diffraction analysis showed that compound **1** crystallized in a cubic space group of *Pn*-3. Relevant crystal data and structural refinement details are summarized in Table 1. The asymmetric unit of compound **1** is shown in Figure 2, including one [Zn(en)]^2+^ ion, two distinct V atoms, three distinct B, and seventeen O atoms. Within the asymmetric unit, the Zn^2+^ ion is in a 6-fold coordination of two N atoms from one ethanediamine ligand and four O atoms from two adjacent borovanadate anions, forming a distorted octahedron geometry. The Zn–N bond length is 2.035(9) Å, and the N–Zn–N angles are 23.7(5)°, which are similar to those in other zinc borovanadates [33]. For the three different B atoms, the B(3) atom is triangularly coordinated by one -OH group and two oxygen atoms, while B(1) and B(2) atoms are tetrahedrally coordinated by four O atoms. The B–O distances vary from 1.353(6) to 1.507(5) Å and O–B–O angles are in the range of 100.6(3)–122.3(4)°. Each V atom is coordinated by four bridging oxygen atoms and one terminal oxygen atom, giving a distorted square pyramidal geometry with V-O bond lengths varying from 1.609(3) to 2.021(3) Å and O–V–O angles from 78.09(12) to 143.72(12)°. Details of the selected bond lengths and angles are given in Appendix A. Bond valence calculations [38,39] revealed that the V atoms were present as an average valence of 4.50, close to a V^IV^:V^V^ ratio of 1:1. The existence of V^IV^ and V^V^ was further confirmed by the X-ray photoelectron spectroscopy (XPS) results, as shown in Appendix A.

Compound **1** exhibited a 3-D open-framework built from [Zn(en)]^2+^ cations and [V_12_B_18_O_54_(OH)_6_(H_2_O)]^10−^ polyanions. In the structure, one trigonal-planar BO_2_(OH) group and two neighboring BO_4_ tetrahedra were joined together to form a {B_3_O_7_(OH)} 3-MR motif. Each {B_3_O_7_(OH)} motif was linked by the adjacent {B_3_O_7_(OH)} motifs via ụ_2_-O atoms to give a puckered {B_18_O_36_(OH)_6_} ring. Six VO_5_ square pyramids were connected with each other, constructing a triangular [V_6_O_18_] unit. Two [V_6_O_18_] units sandwiched one puckered {B_18_O_36_(OH)_6_} ring through ụ_3_-O atoms, leading to one [V_12_B_18_O_54_(OH)_6_(H_2_O)]^10−^ anion with an H_2_O molecule at the center. The sandwich-type borovanadate anion was further linked by six [Zn(en)]^2+^ cations, forming the molecular structural unit of compound **1** (Figure 3). Then, the molecular structural units were extended along the a, b, or c axes into a 3-D open-framework architecture (Figure 4a). Within this 3-D framework, 1-D infinite plane-shaped channels built from four Zn^2+^ cations and four borovanadate anions could be observed, the size of the pore opening was about 11.8 Å × 7.1 Å (Figure 4b).

### 2.3. Catalytic Performance

The oxidation of alcohols to corresponding carbonyls is one of the most important synthetic transformations in organic chemistry with respect to the useful intermediates obtained for the synthesis of fine chemicals [40,41,42,43]. Compound **1**, having V species with mixed oxidation states, may have potential catalytic activity in the oxidation of benzyl-alcohol compounds [44,45]. Thus, we were prompted to investigate the activity of compound **1** in the catalytic oxidation of α-phenethyl alcohol. The optimized reaction conditions were chosen on the basis of the relevant literatures as follows: 1.5 mmol α-phenethyl alcohol as a reaction product, 0.01 mmol compound **1** as a catalyst, 4 ml 30% (wt) H_2_O_2_ solution as the oxidant, 20 mL acetonitrile as the solvent, reaction temperature 75 °C, and reaction time from 1 to 8 h [45,46]. The gas chromatography–mass spectrometry instrument was used to analyze the reaction products. Each yield of the reaction products was determined by the results of multiple parallel experiments. 

GC-MS analysis of the crude reaction mixture showed that acetophenone was the only product of the catalytic reaction that formed in a yield of 45.6% (Appendix A). A control experiment in the absence of compound **1** gave only <6% yield of product under the same conditions, which confirmed that compound **1** plays a significant role in the oxidation of α-phenethyl alcohol. To achieve reaction conditions for maximum yields, the effects of H_2_O_2_ dosage (1,2,4, and 8 mL) on the oxidation of α-phenethyl alcohol to acetophenone were especially investigated. As shown in Figure 5, the yield of acetophenone increased sharply from av. 17.3% to 45.6% when the H_2_O_2_ dosage increased from 1 to 4 mL. Afterward, further increasing the H_2_O_2_ dosage to 8 mL decreased the yield of acetophenone slightly to 45.3%, possibly due to the further oxidation of α-phenethyl to hyperoxide by excess H_2_O_2_. Analogous phenomena have been reported in other literature [46,47,48]. In addition, the catalyst could be easily separated by centrifugation and filtration after completion of the catalytic reaction, and further reused in the successive five-run tests without significant change in the activity (Figure 6). The powder X-ray powder diffraction (XRD) pattern after the oxidation reaction (Appendix A) was collected to estimate the structural stability of **1**. Compared with the powder XRD pattern before the oxidation reaction, no obvious difference was observed, which indicates that the structure of **1** was retained during the catalytic reaction.

As reported earlier, the reaction of benzyl-alcohol with H_2_O_2_/O_2_ used as the oxidant proceeded by means of a radical mechanism [44,46]. According to Okuhara’s opinion [49], the pentavalent vanadium in compound **1** was present in the form of VO_2_^+^ ions, which could rapidly react with H_2_O_2_ to generate large numbers of unstable VO(O_2_)^+^ ions. These VO(O_2_)^+^ ions further reacted with α-phenethyl alcohol and resulted in reactive intermediates that were easily converted into acetophenone.

## 3. Materials and Methods 

### 3.1. Synthesis 

Compound **1** was synthesized from a mixture of NH_4_VO_3_ (1 mmol, 0.117 g), (NH_4_)_2_B_10_O_16_·4H_2_O (6 mmol, 2.83 g), H_3_BO_3_ (6 mmol, 0.371 g), Zn(NO_3_)_2_·6H_2_O (1 mmol, 0.297 g), ethanediamine (10 mmol, 0.671 mL), and H_2_O (5.00 mL) under hydrothermal conditions. The mixture was treated in a 23 mL Teflon-lined stainless-steel autoclave at 180 °C for 7 days. After cooling to room temperature, red block crystals were obtained and gave a yield of 71.3% (based on NH_4_VO_3_). The elemental analysis showed that the mass ratio of C:H:N was 1.73:1.62:2.15, which is consistent with the stoichiometry.

### 3.2. Characterization

X-ray powder diffraction (XRD) was obtained on a Rigaku D/max-2550 diffractometer (Rigaku, Tokyo, Japan) with Cu-K*α* radiation (*λ* = 1.5418 Å). The elemental analysis of the as-synthesized samples was carried out on a Vario MICRO elemental analyzer (PerkinElmer, Waltham, MA, USA). The UV-Vis-NIR diffuse reflectance spectrum was measured at room temperature in the wavelength range of 200–1200 nm using a Shimadzu UV-3600 UV-Vis-NIR spectrophotometer (Shimadzu, Tokyo, Japan). The FT-IR spectrum was recorded on a Bruker-IFS 66 V/S spectrometer (Bruker, Karlsruhe, Germany) using KBr pellets in the 4000–400 cm^−1^ range. The XPS analysis was performed on a VG Scienta R3000 spectrometer (Thermo Nicolet Corporation, Markham, ON, Canada) with a Al-K*a* X-ray source (1486.6 eV). Thermogravimetric analysis (TGA) was conducted in a Perkin-Elmer TG-7 analyzer under nitrogen (PerkinElmer, Waltham, MA, USA). 

### 3.3. Crystal Structure Determination 

Single-crystal X-ray diffraction data for compound **1** was collected at a temperature of 20 ± 2 °C on a Bruker SMART CCD APEX II diffractometer with graphite-monochromated Mo-Kα radiation (*λ* = 0.71073 Å). The structure of compound **1** was solved by the direct method and refined by full-matrix least-squares on *F*^2^ with the SHELXTL crystallographic software package [50]. Solvent molecules could not be located properly due to severe disorder. Consequently, the PLATON/SQUEESE program [51] was used to calculate the disorder area and remove its contribution to the overall intensity data. The V, Zn, B, O, C, and N atoms of the framework were refined with anisotropic displacement parameters. The hydrogen atoms were added geometrically. The number of H_2_O was also comparable by the thermogravimetric analysis and elemental analysis. Crystallographic data for the structure was deposited with the Cambridge Crystallographic Data Centre, CCDC No. 1882299. 

## 4. Conclusions

A new 3-D open-framework zinc borovanadate [Zn_6_(en)_3_][(V^IV^O)_6_(V^V^O)_6_O_6_(B_18_O_36_(OH)_6_)·(H_2_O)]_2_·14H_2_O (**1**) with unique plane-shaped channels was hydrothermally obtained and structurally characterized. Its framework was comprised of trigonal-planar BO_2_(OH) groups, BO_4_ tetrahedra, VO_5_ groups, and ZnO_4_N_2_ polyhedra. Compound **1** exhibited excellent catalytic activities, high recyclability, and stability in the process of catalytic reaction for the oxidation of α-phenethyl alcohol, proving that compound **1** was an effective heterogeneous oxidation precatalyst.

## Figures and Tables

**Figure 1 molecules-24-00531-f001:**
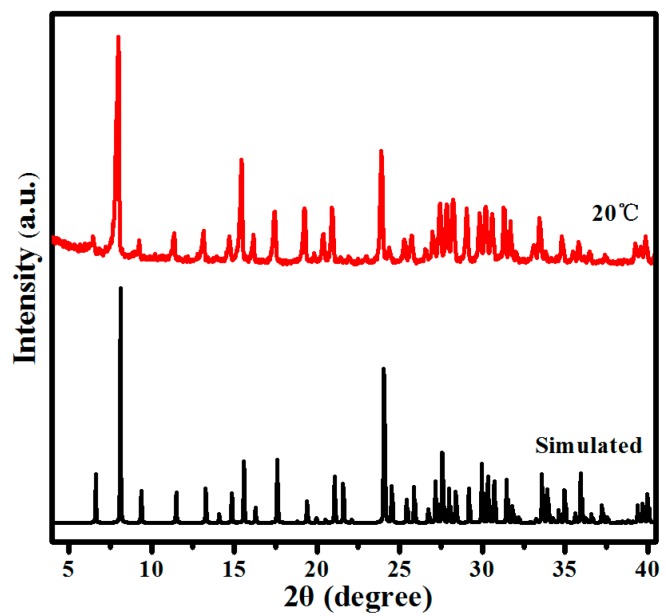
Simulated and experimental X-ray powder diffraction (XRD) patterns of compound **1** in air.

**Figure 2 molecules-24-00531-f002:**
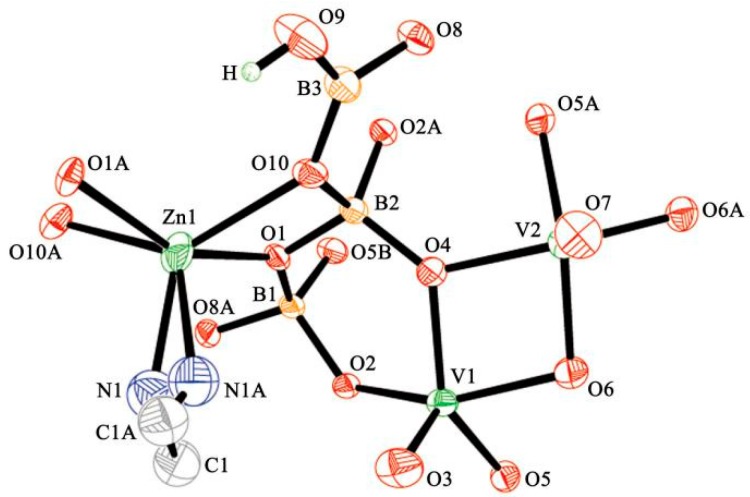
An asymmetric unit of compound **1**.

**Figure 3 molecules-24-00531-f003:**
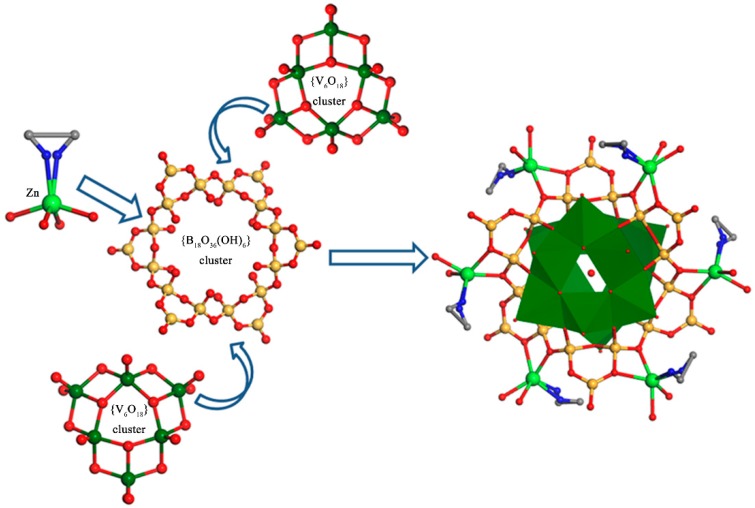
An illustration of the formation of the molecular structural unit of compound **1**. Color code: O, red; V, dark green; B, yellow; Zn, green.

**Figure 4 molecules-24-00531-f004:**
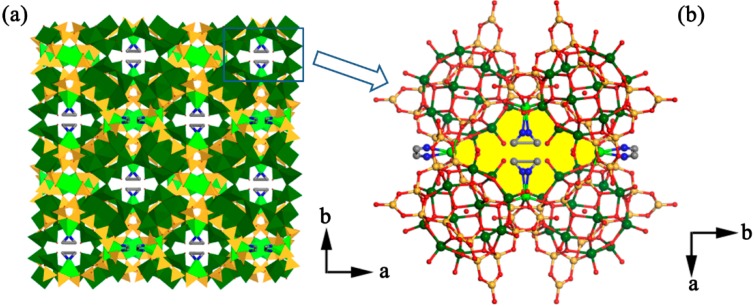
(**a**) View of the 3-D open-framework of **1**; (**b**) view of the plane-shaped channels in **1**. Color code: O, red; V, dark green; B, yellow; Zn, green.

**Figure 5 molecules-24-00531-f005:**
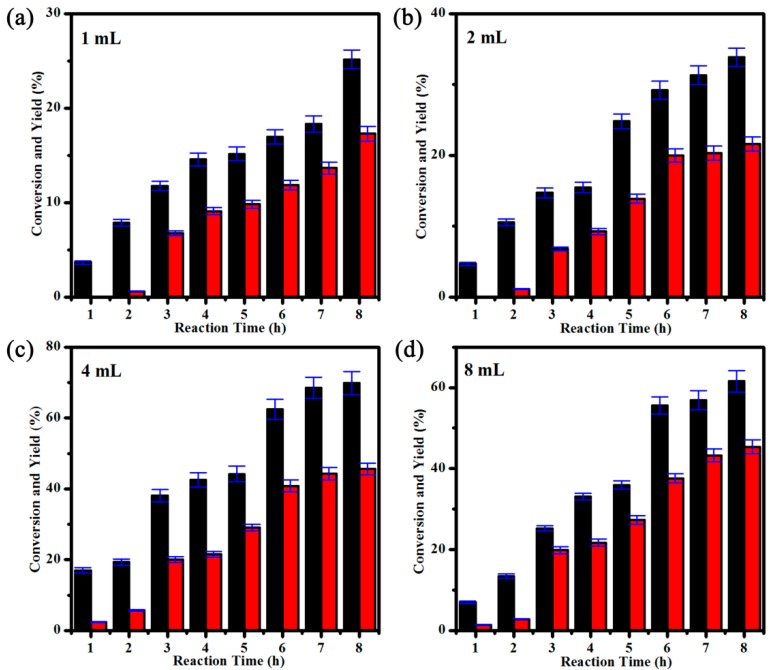
Effect of H_2_O_2_ dosage (**a**) 1 mL; (**b**) 2 mL; (**c**) 4 mL and (**d**) 8 mL on the oxidation of α-phenethyl alcohol (1.5 mmol α-phenethyl alcohol, 0.01 mmol **1**, 20 mL acetonitrile, 75 °C, 8 h). The black bars represent the conversion of α-phenethyl alcohol; the red bars represent the yield of acetophenone.

**Figure 6 molecules-24-00531-f006:**
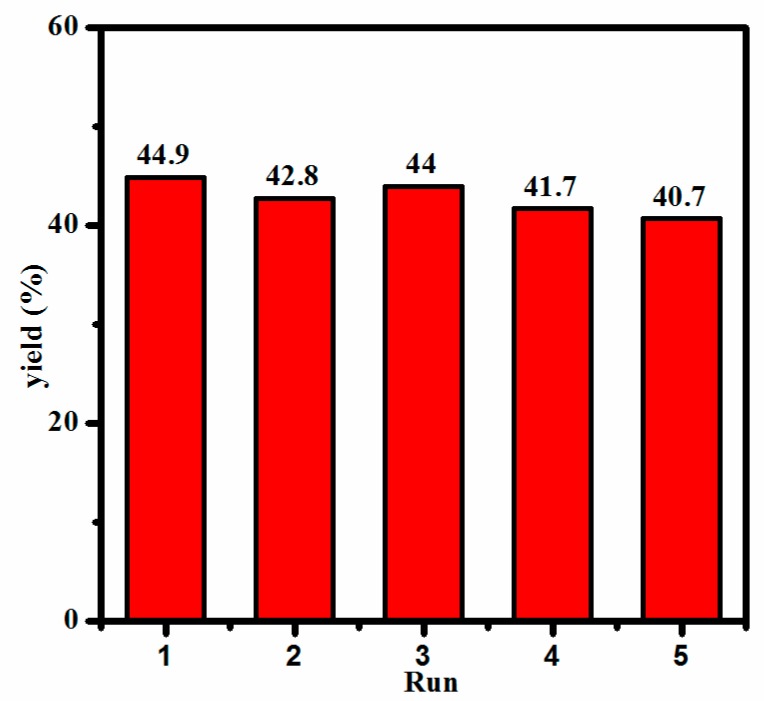
Catalytic oxidation of α-phenethyl alcohol by **1** in different runs.

**Table 1 molecules-24-00531-t001:** Crystallographic data of compound **1**.

[Zn_6_(en)_3_] [V_12_B_18_O_54_(OH)_6_(H_2_O)]_2_·14H_2_O	
Empirical formula	C_6_H_68_B_36_Zn_6_N_6_O_136_V_24_
Formula weight	4404.60
Temperature/K	293(2)
Crystal system	Cubic
Space group	*Pn*-3
*a*/Å	18.850(2)
*b*/Å	18.850(2)
*c*/Å	18.850(2)
*α*/°	90
*β*/°	90
*γ*/°	90
Volume/Å^3^	6698(2)
*Z*	2
Density (calc.)/Mg m^−3^	2.184
Absorption coeff./mm^−1^	2.772
*F*(000)	4292
Crystal size/mm^3^	0.32 × 0.25 × 0.15
2θ range for data collection	6.11 to 54.85°
Index ranges	−24 ≤ *h* ≤ 24 −24 ≤ *k* ≤ 23 −24 ≤ *l* ≤ 24
Reflections collected	63889
Independent reflections	2569[*R*(int) = 0.0917]
Data/restraints/parameters	2569/12/160
Goodness-of-fit on *F*^2^	1.118
Final *R* indexes [*I >=* 2*σ*(*I*)]	*R*_1_ = 0.0473, w*R*_2_ = 0.1138
Final *R* indexes [all data]	*R*_1_ = 0.0576, w*R*_2_ = 0.1197
Largest diff. peak/hole/e Å^−3^	1.323/−1.263

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
