# Peer review of "A New 3-D Open-Framework Zinc Borovanadate with Catalytic Potentials in α-Phenethyl Alcohol Oxidation"

_molecules, 2019, doi:10.3390/molecules24030531_

Round 1

Reviewer 1 Report

Line 52: the amount of ethanediamine is better to write in equivalents

Line 52: Authors should write more about products when the amount of ethanediamine was less or more than 10 mmol.

Line 134: Authors should provide detailed results for screening the optimized reaction conditions in supplementary Information.

Line 140: 45.6 is not good yield! Is it possible to increase the product yield up to 90-95%?

Author Response

Point 1: Line 52: the amount of ethanediamine is better to write in equivalents

Response 1: Thank you very much for the comment. We have modified this sentence and changed it to “When the amount of ethanediamine and NH4VO3 was in a molar ratio of 10 : 1 ”, and the corresponding alterations have also been updated in the revised manuscript.

Point 2: Line 52: Authors should write more about products when the amount of ethanediamine was less or more than 10 mmol.

Response 2: Thank you very much for your advice. We have described the products in detail as below, and the corresponding alterations have been updated in the revised manuscript.

When the amount of ethanediamine was between 0.5 and 5 mmol, only amorphous products were obtained. When the amount of ethanediamine was varied from 6 to 9 mmol, 0-D cage structure [Zn(en)2]6[(VO)12O6B18O39(OH)3]·13H2O was formed, together with black impurities. When the amount of ethanediamine was increased to 10 mmol, pure compound 1 could be synthesized. When the amount of ethanediamine was further increased to 11 mmol, a mixture of compound 1 with poor crystallinity and brown block impurities were obtained.

Point 3: Line 134: Authors should provide detailed results for screening the optimized reaction conditions in supplementary Information.

Response 3: Thank you very much for your advice. In order to obtain the optimal reaction conditions, we have studied the effects of H2O2 dosage and reaction time on the oxidation of α-phenethyl alcohol. Due to the compound 1 has long preparation period and low yield, and this catalytic reaction takes a long time, it is difficult to do a lot of experiments on the effects of solvent type, reaction temperature and amount of catalyst. But we will optimized reaction conditions in the future. We chose this reaction condition based on the relevant literatures, for example, (Dalton Trans., 2015, 44, 8792), (Inorg. Chem. 2011, 50, 9942–9947) and ( Tetrahedron. 2011, 67, 8544-8551).

Point 4: Line 140: 45.6is not good yield! Is it possible to increase the product yield up to 90-95%?

Response 4: Thank you very much for your advice. It’s hard to increase the product yield up to 90-95% at our current level. But, we will do more research and exploration in future experiments.

Reviewer 2 Report

The manuscript submitted by Yuan et al describes the synthesis and structure of a 3D framework based on a vanadyl borate and show some catalytic activity. The science is rather pedestrian and does not add much to the field of catalysis with open framework materials, but the structure might be of some interest to the community, thus it may be published for that reason.

Minor comments.

The authors state that Zn2+ ions are pivotal to structure directing and formation of the 3D framework. Why? is is size or an electronic effect (lack of CFSE)?

Catalysis: There are no error bars on any of the measurements. Why use ml as the units for H2O2 rather than mol%? In Fig 5 what are the significance of the black and red bars?

Author Response

Point 1: The authors state that Zn2+ ions are pivotal to structure directing and formation of the 3D framework. Why? is size or an electronic effect (lack of CFSE)?

Response 1: Thank you very much for your comment. In order to investigate the influence of the distinct metal ions on the structural construction of 3D products, Mn(CH3COO)2·4H2O, Cu(NO3)2·3H2O, and Ni(NO3)2·6H2O were used instead of Zn(NO3)2·6H2O. But, isotypic compounds could not be obtained. This result demonstrated that Zn2+ ions play a pivotal structure directing role in the formation of the open framework. Analogous reports also occurs in other borovanadates, such as in Dalton Trans., 2015, 44, 18731–18736 and Dalton Trans., 2015, 44, 8792. I think both the size and electron effect affect the formation of this 3-D open-framework borovanadate, but the reasons need to be further studied.

Point 2: Catalysis: There are no error bars on any of the measurements. Why use mol as the units for H2O2 rather than mol%? In Fig 5 what are the significance of the black and red bars?

Response 2: Thank you very much for the comment. We have added error bars on the measurements, as depicted in Fig 5 in the revised manuscript. I think using both units of H2O2 can describe the reaction trend. We are sorry for our carelessness.The black bars represent the conversion of α-phenethyl alcohol; the red bars represent the yield of acetophenone. The relative significance has been updated in the revised manuscript.

Reviewer 3 Report

The authors have ignored to reference Zubieta’s almost identical work in 90s with ethylenediamine with almost identical borate clusters VOBOH-1 to 4, Inorganica Chimica ActaVolume 282, Issue 1, 10 November 1998, Pages 123-129. In addition, Zubieta studied the ethylenediamine and longer diamines in porous space formation with vanadophosphonate system Inorg. Chem.200645 (8), pp 3224–3239. Literature review is very poorly conducted. To give another example, reference 4 from 2002 by Clearfield is cited as recent developments in phosphonate chemistry although Clearfield had published the updated version in 2012. 

This work is a very tiny contribution to the previously published work. The catalysis studies are routine and not significant enough to be published at Molecules.

Author Response

Point 1: The authors have ignored to reference Zubieta’s almost identical work in 90s with ethylenediamine with almost identical borate clusters VOBOH-1 to 4, Inorganica Chimica Acta Volume 282, Issue 1, 10 November 1998, Pages 123-129. In addition, Zubieta studied the ethylenediamine and longer diamines in porous space formation with vanadophosphonate system Inorg. Chem., 2006, 45(8), pp 3224–3239. Literature review is very poorly conducted. To give another example, reference 4 from 2002 by Clearfield is cited as recent developments in phosphonate chemistry although Clearfield had published the updated version in 2012.

Response 1: Thank you for the suggestion. I am sorry for my neglecting Zubieta’s contribution in borovanadates. I've found these four borate clusters VOBOH-1 to 4 reported by Jon Zubieta in the literatures, including (enH)(enH2)4[(VO)12B17O38(OH)8]·H2O (VOBOH-1; en=ethylenediamine), (enH2)5[(VO)12O6{B3O6(OH)}6]·H2O (VOBOH-2), (H3O)12[(VO)12{B16O32(OH)4}2]·28H2O (VOBOH-3) and Rb4[(VO)6{B10O16(OH)6}2]·0.5H2O (VOBOH-4). Among these four compounds, VOBOH-1, VOBOH-2 and VOBOH-3 are all built up from discrete VBO polyanions, together with amine counterions or water molecules. Hydrogen bonds align the polyanions, cations or water molecules of these compounds, through which the respective anionic species are integrated into 2D or 3D architectures. While the framework of VOBOH-4 is constructed by discrete [V6B20O38(OH)12]4- clusters with Rb+ counterions floating around. In our work, a novel 3-D open-framework zinc borovanadate [Zn6(en)3][(VO)6(VO)6O6(B18O36(OH)6)·(H2O)]2·14H2O (en = ethylenediamine) has been hydrothermally obtained and structurally characterized. The framework is built from [V12B18O54(OH)6(H2O)]10- polyanion clusters bridged by Zn(en) complex fragments. I think our compound is quite different from the other four, and should be meaningful.

As we known, borovanadates are a growing family among the polyoxometalates due principally to the structural plasticity. Till now, various BVO clusters have been observed in borovanadates, as exemplified by [V6B20O50], [V6B22O53], [V10B28O74], [V12B16O58], [V12B17O58], [V12B18O60], and [V12B32O84]. These clusters are generally interacted with or bonded to different cationic species such as protonated amines, ammonium and hydronium ions, transition metals, and lanthanide ions, thus increasing the dimensionality of these systems from 0D to 1D, 2D or 3D. Although so many extended borovanadates were reported, relatively little progress has been made on the preparation of the three-dimensional open-framework borovanadates. Moreover, few investigation of such borovanadates on the capability and application has been conducted up to now. Therefore, the design and synthesis of 3D borovanadates with novel properties are a promising and meaningful work.

We thank again for the reviewer’s comments. We have modified the introduction carefully and the relevant references have been updated in the revised manuscript.

Round 2

Reviewer 1 Report

The previous version of the manuscript contained some serious incorrect conclusions and inadequate expressions, but now, in the present version, most of them have been revised or omitted. This reviewer considers that the manuscript is acceptable for publication after minor revisions listed below.

Add reference 

for MOF DOI: 10.1039/C5NJ02411K, DOI: 10.1039/c8cc07734g

for alcohol oxidation DOI: 10.1021/ic501704g, DOI: 10.1039/C5RA02667A

Author Response

Point 1: The previous version of the manuscript contained some serious incorrect conclusions and inadequate expressions, but now, in the present version, most of them have been revised or omitted. This reviewer considers that the manuscript is acceptable for publication after minor revisions listed below.

Add reference

for MOF DOI: 10.1039/C5NJ02411K, DOI: 10.1039/c8cc07734g

for alcohol oxidation DOI: 10.1021/ic501704g, DOI: 10.1039/C5RA02667A

Response 1: Thanks a lot for the comment. We have added the references accordingly in the revised manuscript.

Reviewer 3 Report

The authors have improved the text compared to the previous one. Now, it will be acceptable with minor changes. For example, in addition to the changed reference number 4 "Clearfield, A. Chapter 1: The Early History and Growth of Metal Phosphonate Chemistry. 2012." The most recent review articles on this topic pulished in 2018 at "Coordination Chemistry Reviews, Volume 369, 15 August 2018, Pages 105-122 " and  at "European Journal of Inorganic Chemistry 2016 (27), 4300-4309" should be cited as well to be up to date. 

It is also a nice idea to include a table to make the comparison with the previosly published cyrstal structures in the same family of compounds. This will be very helpful for people interested in this field.

Author Response

Point 1: The authors have improved the text compared to the previous one. Now, it will be acceptable with minor changes. For example, in addition to the changed reference number 4 "Clearfield, A. Chapter 1: The Early History and Growth of Metal Phosphonate Chemistry. 2012." The most recent review articles on this topic pulished in 2018 at "Coordination Chemistry Reviews, Volume 369, 15 August 2018, Pages 105-122 " and at "European Journal of Inorganic Chemistry 2016 (27), 4300-4309" should be cited as well to be up to date.

Response 1: Thanks a lot for the comment. The relevant references have been added to the revised manuscript.

Point 2: It is also a nice idea to include a table to make the comparison with the previosly published cyrstal structures in the same family of compounds. This will be very helpful for people interested in this field.

Response 2: Thanks a lot for the comment. We have added a table to make the comparison with the previosly published cyrstal structures in the same family of compounds,as depicted in Table S1 in the revised manuscript.

Table S1. Comparison of cyrstal structure of compound 1 obtained in this study with other reprehensive reports on three-dimensional open-framework borovanadates

Compound

borovanadate   cluster

bridges

Ref.

SUT-6-Zn

(VO)12O6B18O36(OH)6

ZnO5

[17]

SUT-6-Mn

(VO)12O6B18O36(OH)6

MnO6

[17]

SUT-6-Ni

(VO)12O6B18O36(OH)6

NiO6

[17]

SUT-7-Zn

(V10B28O74H8)8-

ZnO5

[7]

{[Cu(dien)(H2O)]3V12B18O54(OH)6(H2O)}·4H3O·5.5H2O

(VO)12O6B18O36(OH)6

Cu(en)2

[32]

[CuII(en)2]4{Na(H2O)(μ-OH)[B(OH)2]}2[(VVO)2(VIVO)10O6(B18O36(OH)6)]}·7H2O

(VO)12O6B18O36(OH)6

Cu(en)2,

Na(H2O)(μ-OH)[B(OH)2]

[36]

[Cd3(H2O)6][(VO)6(VO)6O6)(B18O36(OH)6)]·10H2O

(VO)12O6B18O36(OH)6

Cd(H2O)2O4  

[45]

[Zn6(en)3][(VO)6(VO)6O6(B18O36(OH)6)·(H2O)]2·14H2O

(VO)12O6B18O36(OH)6

Zn(en)

In this work
